# A Beta/ZSM-22 Zeolites-Based-Mixed Matrix Solid-Phase Dispersion Method for the Simultaneous Extraction and Determination of Eight Compounds with Different Polarities in *Viticis Fructus* by High-Performance Liquid Chromatography

**DOI:** 10.3390/molecules24193423

**Published:** 2019-09-20

**Authors:** Gaogao He, Jin Li, Xiaoli Pang, Hui Wang, Hua Jin, Jun He, Shi-Ming Fang, Yan-Xu Chang

**Affiliations:** 1Tianjin State Key Laboratory of Modern Chinese Medicine, Tianjin University of Traditional Chinese Medicine, Tianjin 301617, China; hggtjutcm@163.com (G.H.); zishan826@126.com (J.L.); tgwanghui@163.com (H.W.); zyxyjin@163.com (H.J.); hejun673@163.com (J.H.); 2Tianjin Key Laboratory of Phytochemistry and Pharmaceutical Analysis, Tianjin University of Traditional Chinese Medicine, Tianjin 301617, China; 3Academy of Nursing, Tianjin University of Traditional Chinese Medicine, Tianjin 301617, China; pangxiaoli@163.com

**Keywords:** Beta/ZSM-22 zeolites-based-mixed matrix solid-phase dispersion, flavonoids, HPLC, iridoid glycosides, phenolic acids, vanillin, *Viticis Fructus*

## Abstract

*Viticis Fructus* (VF) was named Manjingzi as a commonly used traditional Chinese medicine (TCM) targeting various pains and inflammation for more than 2000 years. To guarantee the quality of *Viticis Fructus*, a simple, quick and eco-friendly Beta/ZSM-22 zeolites-based-mixed matrix solid-phase dispersion method (B/Z-MMSPD) was established for simultaneous extraction and determination of eight compounds (two phenolic acids, two iridoid glycosides, vanillin and three flavonoids) with different polarities from Viticis Fructus by high performance liquid chromatography coupled with a diode array detector (HPLC-DAD). Beta and ZSM-22 were mixed as the sorbent. Water, tetrahydrofuran and methanol were blended with certain ratio as the eluent. Several parameters including types of sorbents, mass ratio of Beta to ZSM-22, mass ratio of matrix to sorbent, grinding time, types, concentration and volume of eluent were optimized. The recoveries of eight analytes were within the range of 95.0%–105% (RSDs ≤ 4.13%). The limits of detection and limits of quantitation ranged from 0.5 to 5.5 μg/g and from 1.5 to 16 μg/g, respectively. Compared to the traditional extract methods, it was a simple, rapid, efficient and green method. The results demonstrated that a simple, rapid, efficient and green B/Z-MMSPD was developed for the simultaneous extraction and determination of eight target analytes with different polarities for quality control of *Viticis Fructus*.

## 1. Introduction

*Viticis Fructus* (VF), the dried ripe fruit of *Vitex trifolia L.* var. *simplicifolia Cham.* or *Vitex trifolia L.* (Verbenaceae), was named Manjingzi as one of commonly used traditional Chinese medicines (TCMs). It has been used as medicine for more than 2000 years in China. It was recorded in the Shennong Ben Cao Jing (Shennong’s Classic of Materia Medica) [1]. It was listed as one of the supreme Chinese herbs for alleviating cold-heat between tendons and bones, damp arthralgia and muscular spasms, eyesight-improving, teeth-strengthening, nine-orifices benefiting, expelling white worms, losing weight and delaying senility [2]. In the other ancient Chinese herbal classic, Qian Jin Yi Fang, VF was used for alleviating headache and promoting hair growth. VF was also adopted to eliminate the symptoms of eye-swelling, eye-itching and eye-drying in Zhou Hou Bei Ji Fang (A Handbook of Prescriptions for Emergencies). According to Ben Cao Gang Mu (Compendium of Materia Medica), the famous classic book of Chinese materia medica, it could delay fatigue [3]. As documented in the Pharmacopoeia of the People’s Republic of China, VF could dispel wind-heat and improve mentality and vision. It has the effect of alleviating pyrexia, eyes and gums pain, giddiness and headache [4,5]. In Japan, VF derived from the fruit of *Vitex rotundifolia* or *Vitex trifolia*, used for sedation, relieving pains and diminishing inflammation. Moreover, its congenic species, *Vitex agnus-castus*, was used as a medicine for curing climacteric disorder and premenstrual syndrome in Europe [6].

There were plenty of studies about curing various pains and diminishing inflammation. Otherwise, some investigations have revealed that VF has the antipyretic, analgesic, antibacterial, anti-tumor, antihypertensive and anti-oxidant activities [7,8,9,10,11,12,13]. The chemistry compositions of VF included volatile oils, flavonoids, iridoids, diterpenoid, alkaloids and steroids [5,14,15]. Among these compounds, protocatechuic acid (PCA), *p*-hydroxybenzoic acid (PHBA), agnuside, 10-*O*-vanilloylaucubin (VA), vanillin, luteolin, 5,3′-dihydroxy-6,7,4′-trimethoxyflavanone (DHTMF) and casticin are the main active compounds, which have been reported to obtain various biological and pharmacological activities, such as antibacterial, apoptotic, anti-cancer, anti-inflammatory, anti-oxidant, antimicrobial, proangiogenic, anti-nociceptive and anti-hyperprolactinemia effects [16,17,18,19,20,21,22,23]. Therefore, it is fairly significant to extract and analyze these compounds for quality control and subsequent investigations of VF.

Methodologies for analyzing the chemical constituents in VF differ in the extraction technique used. The reported methods include maceration, heated reflux extraction (HRE), microwave assisted extraction (MAE), ultrasonic-assisted extraction (UAE), supercritical fluid extraction (SFE), solid phase microextraction (SPME) and Soxhlet extraction [24,25,26,27,28,29,30]. The main drawbacks of those mentioned techniques are that they usually required a quantity of samples and organic reagent (maceration, MAE and Soxhlet), specific instrumentation (MAE and SFE), specific fibers (SPME) or considerable time (maceration, HRE, MAE, UAE, SFE and Soxhlet). Fortunately, matrix solid phase dispersion (MSPD), as a versatile technique that integrates disruption, extraction, fractionation and purification, could overcome those drawbacks [31,32,33,34,35,36,37]. In other words, it could be realized to use less samples, organic reagent and extraction duration without specific instrumentation and fibers than the others. However, MSPD has not been used for simultaneous analysis of multiple components in VF. With regard to the separation and detection, liquid chromatography (LC), gas chromatography coupled to mass spectrometry (GC-MS) and ultra-high-performance liquid chromatography coupled to mass spectrometry (UPLC-MS) have been developed to determine some components in VF [5,24,25,38]. Nevertheless, there was no method for simultaneous determination of hydrophilic (PCA, PHBA, agnuside, VA and vanillin) and lipophilic constituents (luteolin, DHTMF and casticin) in VF.

Owing to the large differences in polarities of constituents in VF and limitation of MSPD, only constituents with similar polarities could be extracted simultaneously by one sorbent. As generally known, the selection of sorbent was a decisive step for MSPD method. An ideal adsorbent should have the following properties, including certain hardness to disrupt the sample architecture, suitable adsorption capacity to ensure adsorption and desorption of target compounds, certain selectivity for clean-up and suitable particle size for material transferring and rapid elution [39,40,41]. To achieve simultaneously the maximum extraction yield of hydrophilic and lipophilic constituents, two or more sorbents-Based-Mixed Matrix Solid-Phase Dispersion Method as alternative sample preparation method was needed to extract the main constituents with different polarities in herbal medicine.

Zeolites, owing to their unmatched catalytic properties such as thermal stability, crystal structure, active centers and intrinsic porosity at molecular scale were widely applied for all kinds of interesting chemical processes [42]. In recent years, zeolites were gradually developed as sorbents due to their excellent adsorption properties [43,44,45]. Beta or ZSM-22 possessed prominent adsorption and selectivity to compounds with different polarities. The powerful adsorption ability of them mainly derived from their intracrystalline mesopores, which could be adjusted by changing the SiO_2_/Al_2_O_3_ ratio of them [44,46].

In this study, a Beta/ZSM-22 zeolites-based-mixed matrix solid-phase dispersion method with HPLC-DAD was firstly developed for simultaneous extraction and quantification of eight compounds with different polarities from VF including two phenolic acids (protocatechuic acid and *p*-hydroxybenzoic acid), two iridoid glycosides (agnuside and 10-*O*-vanilloylaucubin), one benzaldehyde (vanillin) and three flavonoids (luteolin, DHTMF and casticin). The mixed sorbent consisting of Beta and ZSM-22 was adopted for matrix solid-phase dispersion method to extract multiple compounds in VF. The three-phase mixture made of water, tetrahydrofuran and methanol was used as eluent in MSPD method. The essential parameters including types of sorbents, mass ratio of Beta to ZSM-22, mass ratio of matrix to sorbent, grinding time, types, concentration and volume of eluent were studied to obtain optimal extraction yield. Compared to the traditional MSPD method, it was easy to obtain the maximum extraction yield for simultaneous extracting hydrophilic and lipophilic constituents utilizing the mixed sorbents. The proposed mixed matrix solid-phase dispersion method was expected to be beneficial for the extraction and determination of constituents across a great polarity span.

## 2. Materials and Methods

### 2.1. Chemicals and Reagents

Reference standards including protocatechuic acid, *p*-hydroxybenzoic acid and luteolin were obtained from Chengdu Desite Bio-Technology Co., Ltd. (Chengdu, China). The other five reference substances were isolated and purified from VF extract by our laboratory and were identified by IR, HPLC-MS and H-NMR spectroscopy. The purities of all compounds were not less than 98%. Beta (40, SiO_2_/Al_2_O_3_ ratio), Beta (100, SiO_2_/Al_2_O_3_ ratio), ZSM-22 (70, SiO_2_/Al_2_O_3_ ratio), ZSM-35 (60, SiO_2_/Al_2_O_3_ ratio) and ZSM-5 (300 nm) were supplied from Nanjing JI Cang Nano Technology Co., Ltd. (Nanjing, China). Alumina-A was supplied from Welch Materials (Shanghai, China). ODS C18 (50 μm, 60A) was supplied from Agela technologies. HPLC-grade acetonitrile (ACN) and tetrahydrofuran (THF) was obtained from Concord Technology (Tianjin, China). HPLC-grade methanol (MeOH) was obtained from Fisher (Leicestershire, UK). LC-MS/HPLC-grade formic acid (FA) was obtained from Anaqua Chemicals Supply (Wilmington, USA). Other reagents were of analytical grade. Ultrapure water was obtained from a Milli-Q academic ultrapure water system (Millipore, Milford, MA, USA). All reagents were filtered through a 0.22 µm nylon syringe filter for HPLC analysis.

### 2.2. Plant Material

The seven dried ripe fruit samples of VF were separately purchased from Guangxi, Guangdong, Hubei, Sichuan, Shandong, Hebei and Anhui province in China. Those crude materials of VF were smashed into powder by a pulverizer (Zhongcheng Pharmaceutical Machinery) before passing through a 0.355 mm sieve.

### 2.3. Preparation of Standard Solutions

All target analytes except agnuside were individually dissolved in MeOH at the concentration of 1 mg/mL, respectively. Agnuside (2 mg/mL) was dissolved in MeOH. Appropriate amounts of above solutions were accurately pipetted to prepare the mixed stock solution containing 45 µg/mL protocatechuic acid, 46 µg/mL *p*-hydroxybenzoic acid, 146 µg/mL agnuside, 38 µg/mL 10-*O*-vanilloylaucubin, 50 µg/mL vanillin, 10 µg/mL luteolin, 50 µg/mL 5,3′-Dihydroxy-6,7,4′-trimethoxyflavanone and 80 µg/mL casticin and then stored at 4 °C before analysis.

### 2.4. HPLC Analysis

The analyses of all samples were carried out on an Ultimate 3000 HPLC (Thermo Scientific, Waltham, MA, USA) coupled with a diode array detector (DAD). A Syncronis C_18_ column (250 mm × 4.6 mm × 5 μm, Thermo Scientific, USA) was used for chromatographic separation. The column temperature was fixed at 30 °C, and the injection volume was 10 μL. The mobile phase was composed of 0.4% (*v*/*v*) formic acid (A) and acetonitrile (B) with the gradient elution: 15%–17%B (0–6 min), 17%–20%B (6–15.8 min), 20%–25%B (15.8–18 min), 25%B (18–20 min), 25%–43%B (20–38 min), 43%B (38–42 min), 43%–60%B (42–50 min), 60%–100%B (50–52 min) at the flow rate of 1 mL/min. The wavelength of detector was set at 270 nm. Under the above chromatographic conditions, excellent separations were achieved for all target analytes as showed in Figure 1.

### 2.5. B/Z-MMSPD Procedure

The sample (20.0 mg) of VF, 37.5 mg Beta (40, SiO_2_/Al_2_O_3_ ratio) and 22.5 mg ZSM-22 (70, SiO_2_/Al_2_O_3_ ratio) were accurately weighed and blended in an agitated mortar for 75 s. Then the mixture was transported into a 3-mL polypropylene SPE cartridge with a sieve tray loaded at the bottom in advance. The second plate was put subsequently. The eluent was a mixture of water: tetrahydrofuran: methanol (3:3:4, *v*/*v*/*v*). The power of elution was afforded with a vacuum pump. The resulting solution was collected in a 1-mL volumetric flask in which 0.25 mL eluent was already added. The solution was homogenized by vortex for about 10 s and then filtered through a filter membrane (0.22 µm). The filtrate was analyzed by HPLC.

### 2.6. Heating Reflux Extraction

The sample (2.000 g) of VF was accurately weighed and transported into a 100-mL round-bottom flask. After adding 50 mL MeOH, the total weight was recorded. Heated and refluxed for 1 h, the mixture was cooled down and made up to the recorded weight with methanol. After being shaken well, the extraction solution was filtered with a 0.22 µm filter membrane and then injected into HPLC [4].

### 2.7. Ultrasonic-Assisted Extraction

The sample (1.000) of VF was accurately weighed and transferred into a 100-mL Erlenmeyer flask, then blended with 50 mL MeOH. After recording the total weight, the mixture was processed with ultrasonic extraction (40 KHz, 180 W) for 1 h and cooled to room temperature. The loss of total weight was supplemented with MeOH. After shaking well, the solution was filtered with a 0.22 µm filter membrane before HPLC analysis [5].

## 3. Results and Discussion

### 3.1. Optimization of B/Z-MMSPD Method

#### 3.1.1. Type of Sorbent

The suitable sorbents were indispensable for MSPD method in that they determined whether there were high adsorption capacity and selectivity between sorbents and samples. Herein, seven types of adsorbents (Beta (40, SiO_2_/Al_2_O_3_ ratio), Beta (100, SiO_2_/Al_2_O_3_ ratio), ZSM-22 (70, SiO_2_/Al_2_O_3_ ratio), ZSM-35 (60, SiO_2_/Al_2_O_3_ ratio), ZSM-5 (300 nm), Alumina-A and ODS C18) were screened. As exhibited in Figure 1A,B, there was about 43 min difference of retention time between protocatechuic acid and casticin, which demonstrated that their polarities vary greatly. As showed in Figure 2A, it could not be achieved that all compounds obtain the maximum extraction efficiency when only one type of adsorbent was used. It was found that four analytes (protocatechuic acid, agnuside, 10-*O*-vanilloylaucubin and vanillin) obtained the maximum extraction efficiency by Beta (40, SiO_2_/Al_2_O_3_ ratio) and another four analytes (*p*-hydroxybenzoic acid, luteolin, DHTMF and casticin) obtained the maximum extraction efficiency by ZSM-22 (70, SiO_2_/Al_2_O_3_ ratio). Therefore, Beta (40, SiO_2_/Al_2_O_3_ ratio) and ZSM-22 (70, SiO_2_/Al_2_O_3_ ratio) were chosen as the mixed adsorbents.

#### 3.1.2. Mass Ratio of Beta to ZSM-22

Considering the different selectivity and adsorption capacity between Beta and ZSM-22, the mass ratio of them should be regarded as a vital factor to optimize. Thus, the mass ratio of Beta to ZSM-22 (8:0, 7:1, 6:2, 5:3, 4:4, 3:5, 2:6, 1:7, 0:8) with the total amount of sorbents fixed at 60 mg was studied. As showed in Figure 2B, the extraction efficiencies of four analytes (protocatechuic acid, agnuside, 10-*O*-vanilloylaucubin and vanillin) were gradually decreased while those of another four analytes (*p*-hydroxybenzoic acid, luteolin, DHTMF and casticin) were gradually increased with the ratio from 8:0 to 4:4. With the ratio from 4:4 to 0:8, the extraction efficiencies of some target compounds were distinctly decreased. As the dispersing sorbent of MSPD, the mixed zeolites were not only used as an abrasive to break the sample architecture for exposing the sample compounds but also a solid support to disperse the sample for promoting solvent–sample interactions. The mixed pattern could integrate the respective advantages of Beta and ZSM-22 for improving the extraction efficiencies of target compounds. However, the clean-up function of the mixed sorbent was not significant (Figure 1A,B). In order to obtain the maximum total content, the Beta/ZSM-22 zeolites ratio of 5:3 was adopted for the next step of experiments.

#### 3.1.3. Mass Ratio of Matrix to Sorbent

The mass ratio of sample to sorbent was also one of the important factors that could influence the extraction efficiency of the target compounds by affecting the contact area between sample and sorbent. It could be seen that the extraction efficiencies of most compounds were gradually increased with the ratio from 1:0 to 1:3 (Figure 2C). Because of that, more sorbent could lead to the more sorbent–analytes and solvent–sample interface area, which promoted the extraction of target compounds into solvent. Nevertheless, the extraction efficiencies of some analytes were evidently decreased with the ratio from 1:3 to 1:4. The reason was that too much sorbent produced the adsorption for target compounds resulting in difficult elution. Thus, 1:3 was chosen as the optimal value of sample/sorbent ratio.

#### 3.1.4. Grinding Time

Grinding time was another factor to be investigated for MSPD method. Grinding time could influence the mean particle size and particle size distribution of sample and sorbent when they were blended with a pestle in a mortar. The longer the grinding time was, the tinier the particles that were obtained. A series of grinding times (30 s, 45 s, 60 s, 75 s, 90 s and 105 s) were tested. It was found that the contents of nearly all target analytes were increased with grinding time increasing from 30 s to 75 s (Figure 2D). The reason could be that the longer the grinding time was, the tinier the particles that were obtained and the more chances there would be for compounds to transfer from sample to solvent. However, the contents of all target analytes were decreased when grinding time increased from 75 s to 105 s. It was probably the reason that grinding with longer time obtained excessively tiny particles, which led to harder elution [40]. Therefore, 75 s of grinding time was chosen.

#### 3.1.5. The Organic Part of Elution Solvent

The elution solvent was also a crucial factor of MSPD method. Appropriate elution solvent could obtain the maximum elution on not only compound species but also compound amount. Different elution solvents, such as methanol, tetrahydrofuran, ethanol, acetonitrile and ethyl acetate, were investigated. As showed in Figure 3A, all target compounds could be eluted by methanol, and methanol could gain higher contents of them. Although other elution solvents cannot elute all target compounds, tetrahydrofuran could obtain higher content of casticin than methanol because of the lower polarity. In order to achieve the maximum elution efficiency, methanol and tetrahydrofuran were selected as the mixed elution solvent.

The concentration of elution solvent could make a significant impact on elution process. According to our previous research, the low polar components would be hardly eluted when the percentage of water was over 60% (*v*/*v*). Therefore, the mixed solvent composed of water in range of 0%–60% (*v*/*v*), tetrahydrofuran in range of 0%–90% (*v*/*v*) and methanol in range of 10%–100% (*v*/*v*) were investigated by Optimal (custom) Design of Mixture (Design-Expert 10.0.7). In this investigation, nine systems of three phases were studied. The total content of all target compounds was regarded as the response to fitting model of experimental design. As showed in Figure 3B, the result of fitting performed well with only a 0.22% chance that could occur due to noise. However, higher total content was not obtained than that by using system 9 (3H_2_O: 3THF: 4MeOH) when the solutions proposed by the model of experimental design were adopted. Therefore, system 9 was chosen as the optimal concentration of elution solvent.

The volume of eluent determined whether all the target analytes got eluted thoroughly. With far from enough eluent, the compounds adsorbed in sorbent could not be eluted completely, whereas overmuch eluent could lead to the elution of impurities, longer elution time and even loss of some target compounds. Different volumes of eluent were studied, and the result was exhibited in Figure 3C. It could be found that the extraction yields of most target analytes were increased when the volume of eluent increased from 0.25 to 0.75 mL. However, the content of agnuside was declined with the volume from 0.75 to 1.25 mL. Therefore, 0.75 mL was chosen as the volume of eluent.

### 3.2. Method Validation

#### 3.2.1. Selectivity and Linearity

At least six different amounts of mixed stock solution were added into the adsorbents selected in this work and then processed by the MSPD method. After HPLC analysis, the calibration curves were obtained by plotting the peak areas (*Y*-axis) *versus* the concentrations of compounds in µg/mL (*X*-axis). Fitted by a weighted (1/x) least-squares linear regression method, the calibration curves of eight compounds had high correlation coefficients (r > 0.9992) (Table 1).

#### 3.2.2. Limits of Detection and Quantification

The limits of detection (LOD) were recognized as the concentrations of related compounds in which the signal-to-noise (S/N) obtained 3. The limits of quantification (LOQ) were regarded as the concentrations of related compounds when S/N was equal to 10. The LODs and LOQs of eight compounds ranged from 0.5 to 5.5 and 1.5 to 16 μg/g, respectively (Table 1).

#### 3.2.3. Reproducibility

The data of repeatability was obtained from six independent samples processed by the optimized MSPD method. The RSDs of eight compounds were not more than 4.76% (Table 1). These data indicated that the reproducibility of the optimized method performed well.

#### 3.2.4. Precision, Stability and Recovery

The precision of method was assessed by intra-day (within one day) and inter-day (within continuous three days) precision. They were expressed as RSDs from six replicates of mixed standard solutions at three levels of concentrations. The RSDs of precision (intra-day and inter-day) for eight analytes were below 4.2%, while the accuracies of them ranged from 95.0% to 105%, respectively. (Table 2).

The stability of method was evaluated at room temperature over 24 h. The accuracies of eight analytes with three level concentrations were determined. The accuracies of eight analytes ranged from 95.0% to 105% with the RSDs less than 3.5%, which implied that the stability of eight compounds performed well at room temperature for 24 h. (Table 2)

The recoveries for spiked samples were applied to assessing the accuracy of the optimized MSPD method. 10 mg blank VF samples (or 10 mg blank VF samples and certain amounts of mixed standard solution) were added into the weighed adsorbents and then processed by the optimized MSPD method, which were regarded as the unspiked samples (or spiked samples). The contents of eight compounds were calculated by the related calibration curves. As showed in Table 3, the average accuracies of eight analytes were within the range of 95.0%–105% (RSDs ≤ 4.13%).

### 3.3. Application

B/Z-MMSPD method was used to determine the eight target compounds in VF from seven different provinces in China. As exhibited in Table 4, the contents of PCA, PHBA, agnuside, VA, vanillin, luteolin, DHTMF and casticin in VF were in the range of 3.51–21.63, 13.81–64.06, 29.33–184.92, 4.66–12.52, 1.84–5.88, 3.52–8.70, 2.72–24.80 and 8.52–100.77 mg/100g, respectively. However, the contents of VA and DHTMF in VF from Sichuan, vanillin and luteolin in VF from Shandong and DHTMF in VF from Guangdong were below the LOQ. In addition, it was evidently observed that the content of casticin in VF from Guangxi was obviously higher than that in VF from others.

In order to evaluate the extraction efficiency of B/Z-MMSPD method, No.7 sample was used to compare the difference between B/Z-MMSPD method and traditional heating reflux extraction method based on Pharmacopeia of China 2015. As showed in Table 4, the contents of PCA, DHTMF and casticin by B/Z-MMSPD method were higher than those by heating reflux extraction method, while no significant difference was found between the contents of most compounds by these two methods. Furthermore, the developed MSPD method had the advantages of simpler operation, less consumption of sample and solvent, shorter time and higher extraction efficiency than UAE. Therefore, B/Z-MMSPD method was feasible for the extraction of the VF samples for analysis.

### 3.4. Comparison with Other Methods

Several extraction and analysis methods were summarized in Table 5. In terms of extraction methods, B/Z-MMSPD method only required a small amount of samples, organic solvent and extraction time when other methods needed 2–500 g samples, 35–1500 mL organic solvent and 50–1440 min extraction time. For analysis methods, the developed HPLC-DAD method was applied to analyzing eight target compounds within 52 min, while LC-MS method was used for determining seven target analytes within 55 min and demanded more complex parameters optimized. Although the other HPLC methods need short detection time, they were employed to analyze merely one or two of PCA, agnuside, luteolin and casticin. In a word, B/Z-MMSPD method coupled with HPLC-DAD is an uncomplicated, quick, efficient and green method for extracting and analyzing the eight target analytes in VF and its congeneric plants.

## 4. Conclusions

A simple, quick, efficient and eco-friendly Beta/ZSM-22 Zeolites-based-Mixed MSPD method coupled with HPLC-DAD was established for extraction and quantification of eight compounds in VF. Beta and ZSM-22 were mixed as the sorbent for MSPD technique. Water, tetrahydrofuran and methanol were blended with certain ratio as the eluent for MSPD method. It was proven that B/Z-MMSPD method coupled with HPLC-DAD could be applied to extraction and determination of target compounds in VF and its homologous herbal medicines. Mixed adsorbents could be a good option when single sorbent could not extract the main constituents with different polarities in herbal medicine. Multiphase-mixed eluent could be optimized by experimental design software. It was concluded that the proposed Beta/ZSM-22 Zeolites-Based-Mixed Matrix Solid-Phase Dispersion with HPLC-DAD method is an alternative sample preparation and quality evaluation method for herbal medicines containing constituents with different polarities.

## Figures and Tables

**Figure 1 molecules-24-03423-f001:**
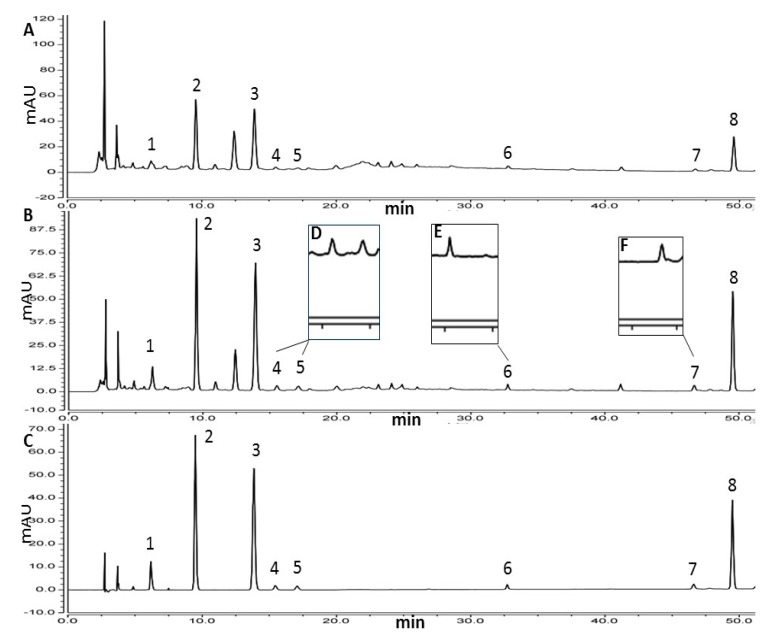
High performance liquid chromatogram of *Viticis Fructus*, extracted by UAE directly with the eluent of MSDP (**A**), extracted by B/Z-MMSPD (**B**), mixture of standard compounds (**C**), double magnification of peak 4, 5 (**D**), 6 (**E**), 7 (**F**). Peaks: 1, protocatechuic acid; 2, *p*-hydroxybenzoic acid; 3, agnuside; 4, 10-*O*-vanilloylaucubin; 5, vanillin; 6, luteolin; 7, 5,3′-Dihydroxy-6,7,4′-trimethoxyflavanone; 8, casticin.

**Figure 2 molecules-24-03423-f002:**
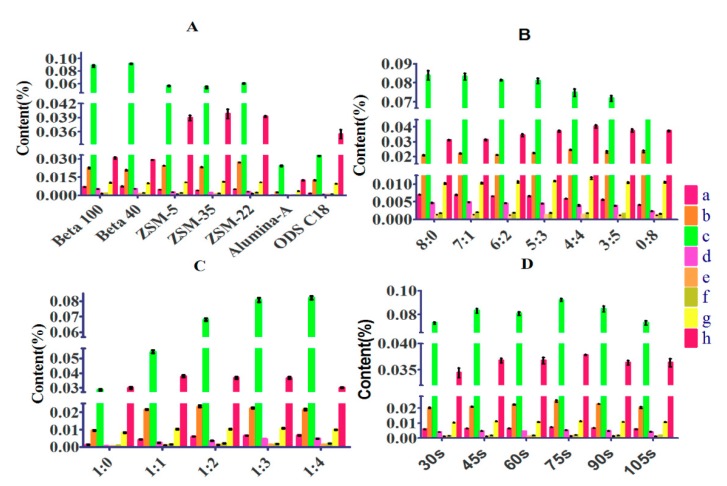
Effects of parameters on efficiency of 8 contents: a, protocatechuic acid; b, *p*-hydroxybenzoic acid; c, agnuside; d, 10-*O*-vanilloylaucubin; e, vanillin; f, luteolin; g, 5,3′-Dihydroxy-6,7,4′-trimethoxyflavanone; h, casticin. (**A**) Types of the sorbent. (**B**) Ratios of Beta to ZSM-22. (**C**) Ratios of sample to sorbent. (**D**) Grinding time.

**Figure 3 molecules-24-03423-f003:**
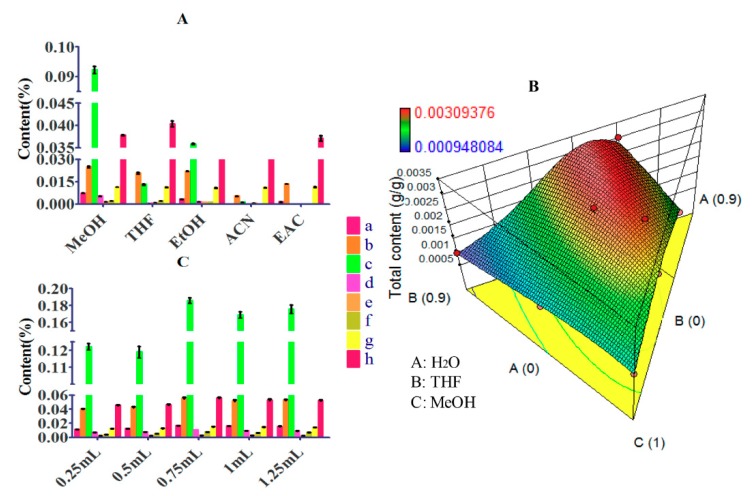
Effects of parameters on efficiency of 8 contents: a, protocatechuic acid; b, *p*-hydroxybenzoic acid; c, agnuside; d, 10-*O*-vanilloylaucubin; e, vanillin; f, luteolin; g, 5,3′-Dihydroxy-6,7,4′-trimethoxyflavanone; h, casticin. (**A**) Eluent type. (**B**) Concentration of elution optimized by Optimal (custom) Design of Mixture (Design-Expert 10.0.7). (**C**) Volume of eluent.

**Table 1 molecules-24-03423-t001:** Linearity, LOD, LOQ and repeatability of the proposed method (*n* = 6).

Compounds	Regressive Equation	Linear Range (µg/g)	r	LOD (µg/g)	LOQ (µg/g)	Repeatability RSD (%)
**PCA**	Y = 0.4676x − 0.0088	9–2250	0.9999	0.5	1.5	2.68
**PHBA**	Y = 0.6944x − 0.0011	9–2300	0.9999	1	4	3.33
**Agnuside**	Y = 0.2439x − 0.0213	29–7300	0.9999	2	7	1.96
**VA**	Y = 0.1547x − 0.0016	7.5–1900	0.9999	1	3	4.54
**Vanillin**	Y = 0.4982x − 0.0033	10–2500	0.9999	1.5	5.5	4.76
**Luteolin**	Y = 0.2433x + 0.0084	10–500	0.9997	2	7	3.57
**DHTMF**	Y = 0.1561x + 0.0095	20–2500	0.9992	5.5	16	2.57
**Casticin**	Y = 0.5268x − 0.0073	16–4000	0.9999	1	2.5	2.79

**Table 2 molecules-24-03423-t002:** The results of precision and stability.

Compounds	Concentration (µg/mL)	Intra-Day		Inter-Day		Stability	
		RSD (%)	Accuracy (%)	RSD (%)	Accuracy (%)	RSD (%)	Accuracy (%)
**PCA**	2	2.0	97.4	1.2	97.4	1.4	96.4
	4	4.2	102.6	2.5	102.6	3.0	101.8
	12	0.0	104.8	0.5	104.9	0.3	104.7
**PHBA**	8.5	1.1	99.2	1.7	99.2	1.8	96.8
	17	2.5	103.8	2.8	103.8	2.6	99.7
	51	0.5	100.8	0.9	101.1	0.6	100.9
**Agnuside**	20	1.8	100.2	2.2	100.2	1.6	97.3
	40	2.8	102.4	3.3	102.4	3.5	100.0
	120	0.7	103.8	0.7	104.0	0.7	104.1
**VA**	1.5	1.0	98.1	1.6	98.1	2.0	98.0
	3	0.4	103.2	0.7	103.2	0.9	103.8
	9	0.1	104.7	0.2	104.6	0.2	104.6
**Vanillin**	0.5	1.4	95.8	1.3	95.8	1.4	96.3
	1	0.3	100.4	0.7	100.4	0.5	101.1
	3	0.1	104.9	0.5	104.2	0.6	104.3
**Luteolin**	0.5	1.3	103.7	1.0	103.7	1.3	104.0
	1	0.4	96.0	0.8	96.0	0.9	95.0
	3	0.2	99.4	0.7	98.6	0.6	98.8
**DHTMF**	1	1.0	100.0	1.8	100.0	1.9	97.7
	2	0.3	98.2	0.4	98.2	0.5	98.0
	6	0.3	95.7	0.7	96.6	0.8	96.5
**Casticin**	7.5	0.7	99.1	1.9	99.1	1.4	96.2
	15	2.7	100.7	2.0	100.7	2.3	99.3
	45	0.7	101.7	1.1	102.9	1.0	102.5

**Table 3 molecules-24-03423-t003:** The results of recovery test (*n* = 6).

Compounds	Unspiked (µg)	Spike (µg)	Spiked (µg)	Average Recovery (%)	RSD (%)
**PCA**	1.49	1.36	2.78	95.0	1.46
**PHBA**	5.70	5.40	10.87	95.8	0.94
**Agnuside**	18.75	14.16	33.06	100	4.13
**VA**	1.41	1.00	2.44	103	0.98
**Vanillin**	0.60	0.30	0.91	103	1.57
**Luteolin**	0.75	0.25	1.00	101	3.22
**DHTMF**	1.39	0.76	2.19	105	1.10
**Casticin**	5.40	5.00	10.27	97.5	0.16

**Table 4 molecules-24-03423-t004:** Contents of the 8 compounds of VF from 7 batches. Average contents (mg/100g) with their standard deviations, n = 3 replicates.

Production Region	PCA	PHBA	Agnuside	VA	Vanillin	Luteolin	DHTMF	Casticin
No.1 (Guangxi)	7.67 ± 0.35	35.13 ± 1.38	98.93 ± 2.25	7.06 ± 0.12	2.77 ± 0.11	8.70 ± 0.03	24.80 ± 1.02	100.77 ± 4.19
No. 2 (Guangdong)	3.51 ± 0.05	13.81 ± 0.21	47.94 ± 2.33	6.91 ± 0.12	1.84 ± 0.01	3.79 ± 0.18	-	30.67 ± 0.55
No. 3 (Hubei)	10.00 ± 0.11	25.12 ± 0.42	73.94 ± 1.75	6.85 ± 0.82	5.83 ± 0.16	5.50 ± 0.26	13.56 ± 0.24	61.36 ± 0.74
No. 4 (Sichuan)	21.63 ± 0.50	53.16 ± 1.00	29.33 ± 1.41	-	5.88 ± 0.12	3.52 ± 0.16	-	29.40 ± 0.51
No. 5 (Shandong)	7.12 ± 0.12	22.34 ± 0.63	34.21 ± 0.50	8.93 ± 0.29	-	-	2.72 ± 0.09	8.52 ± 0.18
No. 6 (Hebei)	6.99 ± 0.24	27.76 ± 1.22	62.66 ± 1.57	4.66 ± 0.04	2.28 ± 0.11	4.60 ± 0.18	12.04 ± 0.29	48.68 ± 2.40
No. 7 (Anhui)	16.49 ± 0.37	64.06 ± 1.79	184.92 ± 6.92	12.52 ± 0.18	4.28 ± 0.17	7.32 ± 0.14	14.78 ± 0.51	55.88 ± 1.71
No. 7 (Anhui) *	10.11 ± 0.25	56.45 ± 0.41	145.72 ± 4.27	10.86 ± 0.30	4.47 ± 0.03	6.22 ± 0.13	13.01 ± 0.13	47.93 ± 0.48
No. 7 (Anhui) **	16.38 ± 0.52	71.01 ± 1.22	201.71 ± 3.18	15.04 ± 0.29	4.65 ± 0.12	9.88 ± 0.16	14.05 ± 0.46	53.60 ± 0.94

* The certain VF sample was extracted by UAE. ** The certain VF sample was extracted by HRE.

**Table 5 molecules-24-03423-t005:** Comparison of the MSPD method with other methods in the determination of compounds in *Fructus Viticis* sample.

No.	Plant	Extracted Compounds	Sample Amounts (g)	Type of Solvent	Solvent Volume (mL)	Extraction Method	Extraction Time (min)	Detection Method	Detection Time (min)	Reference
**1**	*Vitex negundo* and *Vitex trifolia*	p-hydroxybenzoic acid and agnuside	50	Methanol	1500	Maceration	1440	HPLC-PDA	18	[24]
**2**	*Vitex negundo Linn.*	Luteolin	5	Methanol	50	Reflux	120	HPLC	10	[25]
**3**	*Vitex agnus-castus L.* *Vitex trifolia*	Aucubin, homorientin, orientin, agnuside, isovitexin, luteolin-7-*O*-glucoside and casticin	500	Petroleum ether, chloroform and 70%ethanol	-	Maceration	-	LC-MS	55	[38]
**4**	*Vitex trifolia*	Casticin	2	Petroleum ether and methanol	50	Soxhlet	540	HPLC	14	[30]
**5**	*Vitex trifolia*	Luteolin	2.5	MeOH	35	UAE	50	HPLC	20	[27]
**6**		PCA, PHBA, agnuside, VA, vanillin, luteolin, DHTMF and casticin	0.02	A mixture absolute water/tetrahydrofuran/methanol (3:3:4, *v*/*v*/*v*)	0.75	MSPD	1.25	HPLC-DAD	52	This work

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
