# Peer review of "A Beta/ZSM-22 Zeolites-Based-Mixed Matrix Solid-Phase Dispersion Method for the Simultaneous Extraction and Determination of Eight Compounds with Different Polarities in Viticis Fructus by High-Performance Liquid Chromatography"

_molecules, 2019, doi:10.3390/molecules24193423_

Round 1
Reviewer 1 Report
The authors mostly corrected the manuscript in accordance with the suggestions. Explanations of the influence of grinding time was added. The spelling of units was unified.
The authors could still more emphasize the novelty and advantages of their research.
In my opinion, the manuscript may be published in present current form.
Author Response
Response to Reviewer 1 Comments
The authors mostly corrected the manuscript in accordance with the suggestions. Explanations of the influence of grinding time was added. The spelling of units was unified.
Point 1: The authors could still more emphasize the novelty and advantages of their research.
In my opinion, the manuscript may be published in present current form.
Response: Thanks for your good suggestion. The novelty and advantages of our research have been emphasized in the manuscript. Compared to the traditional MSPD method, it was easy to obtain the maximum extraction yield for simultaneous extracting hydrophilic and lipophilic constituents utilizing the mixed sorbents. The proposed mixed matrix solid-phase dispersion method was expected to be beneficial for the extraction and determination of constituents across a great polarity span.
Reviewer 2 Report
In their resubmitted revised manuscript, the authors have addressed all of the concerns raised by the reviewers.
Nevertheless, some minor corrections are recommended:
Page 6, Line 220: Please replace “Longer the grinding time was, more the tiny particles obtained” with “The longer the grinding time was, the more tiny particles were obtained”.
Page 6, Line 222: Please replace “too much” with “too many”.
Author Response
Response to Reviewer 2 Comments
In their resubmitted revised manuscript, the authors have addressed all of the concerns raised by the reviewers.
Point 1: Nevertheless, some minor corrections are recommended:
Page 6, Line 220: Please replace “Longer the grinding time was, more the tiny particles obtained” with “The longer the grinding time was, the more tiny particles were obtained”.
Page 6, Line 222: Please replace “too much” with “too many”.
Response 1: Thanks for your good suggestion. “Longer the grinding time was, more the tiny particles obtained” has been replaced with “The longer the grinding time was, the more tiny particles were obtained”. “too much” has been replaced with “too many”.
Reviewer 3 Report
I do not agree with all answers given by the authors to my comments on the original submission.
I had mentioned that in my opinion the discussion of the MSPD process was somewhat confusing. In my opinion, the revised text in lines 196 to 205 is still unclear and misleading. For example, I am not able to understand statements like "target compounds were gradually extracted from samples to the sorbent" or "When the amount of sorbent was at the high level, the extra target compounds were hardly extracted from samples to the sorbents because of the adsorption equilibrium". In fact (as mentioned in my previous comments), in MSPD the solid sample is (mechanically) mixed with the solid sorbent. There is no need that any "adsorption equilibrium" is established for the target analytes at this point. The target analytes are not necessarily "extracted from samples to the sorbent" at this point. When the solvent is added, the target analytes get dissolved and eluted, whereas the matrix components should remain on the sorbent due to strong interactions with the surface of the sorbent. (In this context another comment to avoid confusion: reviewer #2 stated that the sample preparation procedure includes "solid phase extraction". This may lead to confusion. Solid phase extraction and MSPD are not the same (although of course related), and the fundamental precedures for solid phase extraction and MSPD are somewhat different.)
Hydrophobicity of the sorbent: It is true that in the literature statements about hydrophobicity and ratio of SiO2/Al2O3 can be found. However, both SiO2 and Al2O3 are polar materials (and therefore primarily not hydrophobic). It may be true that this ratio is a reason that the material can be a bit more polar or less polar, but I do not fully believe that in total the material would be truely hydrophobic. A final statement can only be given if experimental evidence for hydrophobicity (such as data from contact angle measurements or related techniques) is available. At the moment I recommend to avoid any text in the manuscript that would give the impression that this material would be a typically hydrophobic material.
Removal of mtrix components: The authors say that the direct treatment by UAE extracts more matrix components, and they refer to Figure 1 A and B. When looking at these chromatograms it turns out that the differences are very minor (!). In so far, these two chromatograms show that the sorbent does not lead to significant clean-up (and obviously in this case a true clean-up is not necessary anyway). I believe that some statements are required in the text to make clear the role of the sorbent.
The discussion of the performances of the 1 mL or 3 mL cartridge is still unclear. As far as I understand the text, in both cases the same amount of sorbent was used - therefore, I cannot see good reasons why the results can be different (besides random errors). At the moment I also cannot see why the 1 mL cartridge must lead to much longer elution times.
Author Response
Response to Reviewer 3 Comments
I do not agree with all answers given by the authors to my comments on the original submission.
Point 1: I had mentioned that in my opinion the discussion of the MSPD process was somewhat confusing. In my opinion, the revised text in lines 196 to 205 is still unclear and misleading. For example, I am not able to understand statements like “target compounds were gradually extracted from samples to the sorbent" or "When the amount of sorbent was at the high level, the extra target compounds were hardly extracted from samples to the sorbents because of the adsorption equilibrium". In fact (as mentioned in my previous comments), in MSPD the solid sample is (mechanically) mixed with the solid sorbent. There is no need that any "adsorption equilibrium" is established for the target analytes at this point. The target analytes are not necessarily "extracted from samples to the sorbent" at this point. When the solvent is added, the target analytes get dissolved and eluted, whereas the matrix components should remain on the sorbent due to strong interactions with the surface of the sorbent. (In this context another comment to avoid confusion: reviewer #2 stated that confusion. Solid phase extraction and MSPD are not the same (although the sample preparation procedure includes "solid phase extraction". This may lead to of course related), and the fundamental procedures for solid phase extraction and MSPD are somewhat different.)
Response 1: Thanks for your valuable and professional suggestion. According to the good suggestion, the related texts have been revised and deleted on Page 6, Lines 208-214.
Point 2: Hydrophobicity of the sorbent: It is true that in the literature statements about hydrophobicity and ratio of SiO2/Al2O3 can be found. However, both SiO2 and Al2O3 are polar materials (and therefore primarily not hydrophobic). It may be true that this ratio is a reason that the material can be a bit more polar or less polar, but I do not fully believe that in total the material would be truly hydrophobic. A final statement can only be given if experimental evidence for hydrophobicity (such as data from contact angle measurements or related techniques) is available. At the moment I recommend to avoid any text in the manuscript that would give the impression that this material would be a typically hydrophobic material.
Response 2: Thanks for your valuable and professional suggestion. The hydrophobicity of the sorbent could not be verified only by the literature. In order to avoid confusion, the related texts have been deleted.
Point 3: Removal of matrix components: The authors say that the direct treatment by UAE extracts more matrix components, and they refer to Figure 1 A and B. When looking at these chromatograms it turns out that the differences are very minor (!). In so far, these two chromatograms show that the sorbent does not lead to significant clean-up (and obviously in this case a true clean-up is not necessary anyway). I believe that some statements are required in the text to make clear the role of the sorbent.
Response 3: Thanks for your valuable and professional suggestion. The related statements about the role of the sorbent have been added into the manuscript. (Page 6, Lines 208-214)
Point 4: The discussion of the performances of the 1 mL or 3 mL cartridge is still unclear. As far as I understand the text, in both cases the same amount of sorbent was used - therefore, I cannot see good reasons why the results can be different (besides random errors). At the moment I also cannot see why the 1 mL cartridge must lead to much longer elution times.
Response 4: Thank you for asking such an interesting and professional question. According to the experimental results, it would lead to much longer elution time when the 1 mL cartridge was used. It may be that the smaller radius makes its channel easier to be blocked. Meanwhile, longer contact time between solution and sample would increase the dissolution of sample compounds. However, the real reason needs to be further clarified. In order to avoid confusion, the related texts have been deleted.
Round 2
Reviewer 3 Report
The final revision looks fine.
This manuscript is a resubmission of an earlier submission. The following is a list of the peer review reports and author responses from that submission.
Round 1
Reviewer 1 Report
I have serious doubts that the manuscript is suited for publication.
In my opinion, the authors discuss MSPD in a somewhat confusing way. MSPD is a technique where the solid sample is mixed with a solid sorbent, and the mixture is put into a small cartridge. The sample components are distributed across the surface of the solid sorbent, and the analytes are extracted/eluted with a proper solvent. Ideally the matrix components remain on the sorbent. If there is just a very small interaction of the analytes with the sorbent surface, the analytes will be easily eluted and the extraction efficiency is high. Therefore,the following statement (and similar statements throughout the manuscript) are misleading: "... the extraction yield of each compound was not increased, even some of them were distinctly decreased. It could be explained that the adsorption of B-Analytes was decreased...." (lines 195-197). If the adsorption is decreased, it would be easier to elute the analytes and the extraction yield would increase!
Line 87 and line 180: I am wondering why the authors can assume that their sorbents would be hydrophobic.
Figure 1: It would be necessary to give the numbers for the time on the x-axis, and the numbers for the absorbance units on the y-axis.
In the Introduction the authors say that techniques like HRE or UAE require large quantities of samples and organic reagents. It may be true that the HRE and UAE procedure mentioned in parts 2.6 and 2.7 require large amounts, but it would be easy to miniaturize them and to used much lower quantities of sample and solvent. In so far, HRE and UAE do not necessarily suffer from the mentioned disadvantages.
Figure 1: This is a chromatogram of a real sample. Some of the peaks are okay, but the peaks for analytes 4 and 5 are too small to allow a reliable quantitation. There are obviously small matrix peaks that may or may not interfere.
Part 3.1.5: The header "Elution Solvent" is misleading, because the text discusses only the organic part of the elution solvent, but not the whole elution solvent. In the end the authors use a mixture of water, THF and MeOH, but water is of course also an elution solvent.
Table 1: Linearity, limits of detection an quantification: The data are given as µg mL-1.However, the sample is a solid sample, and for a reader it would be interesting to know such data as µg/g sample.
In general it is unclear how efficiently the extraction procedure gets rid of matrix components. It might be worth trying to extract the samples directly with the eluent of MSDP using for example UAE (maybe at elevated temperature). This would give some information if the MSDP removes matrix components or not.
Lines 303 to 305: For a reader it is very unclear why this 1 mL SPE cartridge works better than the 3 mL cartridge preferred by the authors. According to Table 4, some analytes show significantly higher results with the 1 mL cartridge !! This would mean that one should NOT use the 3 mL cartridge.
Reviewer 2 Report
This manuscript describes the development of a method for the analysis and identification of selected secondary plant metabolites of the dried fruit of the tropical plant Vitex trifolia, termed Viticis Fructus, used in Traditional Chinese Medicine, based on an extraction with zeolites followed by analysis with high performance liquid chromatography. A key part of the method is the sample preparation, which includes solid phase extraction with a mixture of zeolites with an optimized quantity/composition of extraction solvent. This was followed by a method validation in terms of precision, accuracy, recovery and limit of detection/quantitation using eight reference compounds. Finally, the method was developed into a semi-quantitative analysis of various Viticis Fructus samples from a variety of sources in order to provide a fingerprint of these compounds and to estimate the concentration levels.
The manuscript is reasonably well written, with a good introduction and appropriate experimental design. The results are clearly presented with the conclusions supported by the results. However, before a publication in Molecules can be considered, it is recommended to describe the materials, methodologies and results more clearly to enhance the readability of manuscript and to correct typographical errors.
Comment 1: The code names “Beta/ZSM-22” do not mean anything to the reader. Thus, for greater clarity in title, abstract, keywords and body of the manuscript “Beta/ZSM-22” needs to be replaced by “Beta/ZSM-22 zeolites” to indicate the chemical nature of the solid-phase extraction material.
Comment 2: It is unclear what “50-mesh sieve” on Page 3, Line 119 stands for since the unit is missing.
Comment 3: It is unclear in the Legend of Figure 1, whether the sample associated with chromatogram A had undergone a particular solid-phase extraction process or was derived from liquid extraction and directly injected into the HPLC. Also missing from the legend is a brief description for the insert in Panel A for peaks 4 to 7, mentioning the level of magnification. The magnified trace of peaks 4 and 5 in Panel A appear to be offset to higher retention times, which needs to be corrected.
Comment 4: In the material and method section on Page 4, Line 146 it was described that the zeolites were ground with a pestle in a mortar, but it was not mentioned whether the resulting particles were subsequently sieved. Was a particle size measurement performed?
Comment 5: It is suggested to define the terms “B-Analytes” and “Z-Analytes” used on Page 5, Lines 196-197.
Comment 6: It is unclear in the section of “3.1.4. Grinding Time” on Page 6, Lines 210-218 what material is actual ground, the zeolites or the Viticis Fructus sample. This needs to be specified. What consequence does the grinding time have on the sample with respect to particle size?
Comment 7: The use of the nomenclature and English language could be improved e.g.:
Page 2, Line 56: Please replace “has” with “have”.
Page 2, Line 78: Please replace “As known to all” with “As generally known”.
Page 2, Line 86: Please replace “processed” with “possessed”.
Page 3, Line 105: Please replace “H-NMR” with “H-NMR spectroscopy”.
Page 3, Line 131: Please replace “Thermo scientific” with “Thermo Scientific”.
Page 4, Line 139: Please replace “chromatography figure” with “chromatogram”.
Page 4, Line 144: Please replace “Bate” with “beta”.
Page 4, Line 145: Please replace “agate” with “agitated”.
Page 4, Line 146: Please delete “slightly”.
Page 4, Line 147: Please delete “solvent”.
Page 4, Line 148: Please replace “offered by” with “afforded with”.
Page 6, Line 206 and throughout the manuscript: Please replace “overmuch” with “too much”.
Page 6, Line 210: Please replace “not to be ignored” with “to be investigated”.
Page 6, Line 211: Please replace “would benefit to enough contact” with “would allow sufficient contact”.
Page 6, Line 216: Please replace “overlong” with “longer”.
Page 6, Line 221 and throughout the manuscript: Please replace “compound contents” with “compound amount”.
Page 7, Line 236: Please replace “percent” with “percentage”.
Page 7, Line 257: Please italicize “versus”
Page 7, Line 259: Please replace “surely got good” with “high”.
Page 11, Line 2316: Please replace “was only in need of” with “only required”.
References: Please italicize species names and provide correct subscripts in chemical formulas.
Page 13, Line 462: Please replace “……” with relevant compound names.
Reviewer 3 Report
The Authors described the method of simultaneous mixed matrix solid-phase dispersion extraction and determination by HPLC-DAD of eight various compounds in Viticis Fructus. The development of new extraction procedures for analysis of biologically active compounds in plant material is still an important aspect of scientific research. The Authors examined different conditions of extraction procedure such as type of adsorbent, mass ratio of sample to sorbent, grinding time, elution solvent. The method was validated. However, the novelty of the manuscript is not very high. The method for extraction of eight compounds in only one plant raw material was optimized and validated. However, the Authors, proved the advantages of the developed method compared to previously published. Therefore, in my opinion, the manuscript is worth publishing after minor revision.
The Authors should be emphasized the novelty and advantages of the method developed by them.
Explanations of the influence of grinding time is unclear and not supported by results (lines 216-218) and should be corrected.
The spelling of units should be unified, e.g.line 123 is: μg•mL-1, Table 1 is: µg/mL